# Uterine Tube Resection by Vaginotomy as an Alternative to Ovariectomy in Mature Cattle

**DOI:** 10.3390/ani13061066

**Published:** 2023-03-15

**Authors:** Peter C. Irons, Bryce Mooring, Natalie Warburton, Emma Dunston-Clarke, Gavin Pensini, Shona Hay, Teresa Collins

**Affiliations:** 1School of Veterinary Medicine, Murdoch University, Murdoch 6150, Australia; 2Broome Cattle Vets, Djugun 6725, Australia; 3West Coast Veterinary Hospital, Forrestdale 6112, Australia; 4Tableland Veterinary Service, Malanda 4885, Australia

**Keywords:** bovine, surgical spay, gestation, reproductive anatomy, reproductive control, oviduct

## Abstract

**Simple Summary:**

Female cattle should not be in an advanced state of pregnancy when transported from a farm for sale or slaughter, particularly when they are shipped or trucked over long distances. It is not always possible for the farmers in the northern Australian rangelands to prevent access to bulls given the minimal fencing and extreme farming conditions, and there are no practicable non-surgical contraceptive methods. This article describes the use of a little-known surgical technique of removing part of the uterine tubes (also known as the oviducts, the equivalent of the Fallopian tubes in humans) on 70 Brahman cows in a typical farm environment in northern Australia in comparison with 60 Brahman heifers spayed using an established surgical method. The animals were monitored for 10 days following the surgery. None of the animals died following the surgery, 2 of the 70 were treated for illnesses and recovered, and a number of animals subjected to both surgical procedures showed signs of discomfort and mild inflammation in the pelvic area during recovery. The cows gained an average of 9.3 kg of weight over the 10 days of the study compared to the spayed heifers, which lost 3.5 kg. The uterine tube resection technique therefore appears to be a viable alternative to surgical spaying.

**Abstract:**

The prevention of pregnancy is desirable for female cattle destined for sale in the northern Australian rangelands for both economic and welfare reasons. Controlled access to bulls is often not feasible, nor are any non-surgical methods currently available. Many females are therefore surgically spayed. This study describes a technique for uterine tube resection (UTR), which leaves the ovaries intact and is performed using a vaginal approach, and compares the outcomes from 70 Brahman cows subjected to the procedure with 60 heifers spayed using the dropped ovary technique. The animals were monitored for 10 days following the surgery. There were no mortalities, and two animals were treated for illnesses after the UTRs and recovered. The animals subjected to both surgical procedures showed signs of pelvic discomfort and mild inflammation during recovery. The cows gained an average of 9.3 kg (SD 14.5 kg) of weight over the 10 days of the study compared to the spayed heifers, which lost 3.5 kg (SD 13.3 kg), with 19 and 63% of the animals in each of the groups losing weight, respectively. Uterine tube resection can be considered as a viable alternative to surgical spaying.

## 1. Introduction

The rangelands of northern Australia support extensive low-input beef production systems with year-round breeding due to the low labour availability and seasonal rainfall impeding access to cattle [1,2]. The infrastructure is limited and cattle run in large breeder herds, which are mustered once or twice annually to remove and process weaners and cull animals. The animals are typically marketed directly off the property of origin, with the main destinations being for domestic slaughter or live export as feeder or slaughter cattle. Breeding is not controlled in that fertile bulls have access to females at all times. The animals selected for removal from the herd may be pregnant or are at risk of becoming pregnant prior to marketing [3].

The prevention of pregnancy is desirable in the animals destined for market, particularly during the harsh dry season [4]. The non-pregnant animals are thought to gain condition more efficiently without the demands of pregnancy or lactation, although this has not been noted for heifers [5,6,7]. The animals exported live for slaughter or finishing in the overseas markets must be spayed or certified as non-pregnant [8]. Pregnancy also represents a loss in animals destined for slaughter, and is a welfare concern for those transported over long distances, despite current regulations [9].

There are no non-surgical methods of preventing pregnancy available and feasible for use in this production system, although this is an area of research. For this reason, many surplus and culled females are surgically spayed to prevent pregnancy, the economic feasibility of which is supported by modelling [7]. The predominant method for surgical spaying is the dropped ovary technique [10,11,12,13,14]. This is commonly performed by veterinarians and non-veterinary licenced operators in jurisdictions where permitted. While spay procedures have been shown to induce pain, stress, morbidity and mortality, they are mostly conducted without the use of analgesia or anti-inflammatory medication [15]. Thus, practical pain mitigation strategies are in need of development.

Uterine tube resection (UTR), commonly known as ‘webbing’ in Australia, is known to have been conducted since at least the late 1800s when it was executed through a flank incision [16]. More recently, UTR using a vaginal approach has been pioneered and is gaining in popularity. The adoption of this method in preference to other spay techniques is supported by welfare standards for mature cattle [17]. Uterine tube resection is indicated in mature cows destined for removal from the herd [14]. Non-pregnant animals requiring conditioning prior to marketing and pregnant animals up to seven months in gestation are suitable candidates. Only cows of a good size and which have calved previously are considered for the procedure due to the need for the vagina to accommodate the operator’s hand.

No detailed descriptions of the UTR technique are widely available. The aims of this study were to document the surgical technique and the health and performance of a group of animals, and to compare this to animals spayed using the dropped ovary technique under similar conditions.

## 2. Materials and Methods

This study was reviewed and approved by the Animal Research Ethics Committee of Murdoch University.

Performance of the surgical procedures by an experienced practitioner and subsequent observation of animal health and weight change was conducted on a property in Western Australia’s Kimberley region. The property consists of 381,618 hectares of open rangeland where cattle are mustered seasonally and selected for sale on the domestic and export market.

Animals for this study were mustered from rangelands into smaller paddocks four months earlier and moved into yards with ad libitum access to water and hay for two days preceding the study.

A total of 78 Brahman cows and 68 Brahman heifers were used in this study. Heifers and cows were processed similarly with the exception of the surgical procedure, with heifers being spayed using the transvaginal dropped ovary technique while the cows were subjected to uterine tube resection (UTR). UTR was preferred in mature animals in preference to spaying due to their larger ovaries and the associated higher risk of fatal haemorrhage following spaying [14].

Each animal was run up the race, restrained in a crush (Warwick Cattle Crush Company, Manual Exotic Crush, Forest Hill, Australia) with inbuilt electronic pressure plate scales, with the scale being checked the morning of each treatment day using a known weight.

The identity and weight of each animal was recorded, and pregnancy diagnosis was conducted using rectal palpation. Animals were selected by farm staff based on pregnancy status of being either not pregnant or <6 months in gestation and visual characteristics (temperament, body condition, coat colour and presence of horns). A total of 8 heifers and 8 cows were designated as controls and were not subjected to surgery.

The procedures consisted of surgery as described in detail below, immediately followed by restraint of the head in the neck clamp, administration of meloxicam either into the buccal cavity (Ilium Buccalgesic OTM, Troy Animal Healthcare, Glendenning, Australia) or via subcutaneous injection (Ilium Meloxicam, Troy Animal Healthcare) at the manufacturers’ recommended dosages, and identification using coloured chalk and ear-tagging. The surgically sterilised animals were permanent marked with an ear punch. The times of the start of the rectal palpation and the start and finish of the surgical procedure were recorded for each animal.

These procedures were performed over 2 days, after which animals were held in group pens with ad libitum access to water and hay and observed daily by the researchers for signs of poor health for 10 days. Heifers and cows were segregated for the first 24 h, after which they were placed in a single large pen, and then released into an adjoining paddock. Any animals showing signs of illness not consistent with normal recovery during this time were examined clinically and treated as indicated.

Weights were recorded on day 10, being 9- or 10-days post-procedure, which was the end of the study.

The UTR surgical procedure is described in detail and the dropped ovary technique is summarised below.

### 2.1. Surgical Anatomy

The arrangement of the ovaries and their associated ligaments are distinctive and must be well understood for successful location and manipulation. They are described and depicted in anatomical texts [18,19,20], and are summarised here with reference to the procedure. The English versions of the terminology in the World Association of Veterinary Anatomists Nomina Anatomica Veterinaria (2017) are used in this description.

The external ostium of the cervix is surrounded by the fornix, the shallow blind-ending cranial part of the vaginal cavity. Dorsal to the cervix, the fornix lies against the caudal pouch of the peritoneum, which extends caudally between the rectum above and the cervix below [12]. This direct apposition and the location of the ovaries close to the uterine body at all stages except late gestation facilitate access to the ovaries via the vaginal fornix.

The ligamentous attachments of the ovaries allow for a large variation in positioning relative to the pelvis. They generally lie in the vicinity of the pelvic brim, craniolateral to the cervix and adjacent to the uterine body. They are entirely intrapelvic in the non-pregnant state and lie well below the pelvic brim in advanced pregnancy.

Regardless of the variation in relation to the pelvis, the ovaries are held close to the uterine body and tips of the uterine horns by the mesovarium (Figure 1), which is the part of the broad ligament supporting the ovaries. The mesovarium adjoins the mesometrium, which attaches to the uterine body and horns. The ligamentum ovarium proprium runs in the mesometrium and attaches the ligamentous pole of the ovary to the uterus.

The mesosalpinx is a separate fold of the broad ligament holding the uterine tube, which arises at the tubal pole of the ovary and runs the length of the uterine tube, supporting it between the ovary and the uterine horn. The ovarian bursa is a blind-ending pouch between the mesovarium and the mesosalpinx, lying ventrolateral to the ovary. The bursa can be felt during rectal palpation by holding the ovary between the fingers and thumb and extending the fingers down the mesovarium in a cranioventral direction such that one or more fingertips enter and are trapped in the bursa (Figure 1). This is the same principle applied in this surgical technique.

The uterine tube starts at the infundibulum, which lies on the edge of the ovarian bursa close to the tubal pole of the ovary. The tube runs the length of the bursa to the tip of the uterine horn, which is close to the ligamentous pole of the ovary. It is 2–3 mm in diameter in an adult cow. Although soft in consistency, its location can be determined accurately, even if it is not palpable, by the location in the mesosalpinx, which can generally be clearly felt. The close apposition of these structures results in the tube being transected if a segment of the mesosalpinx is removed.

During the procedure, care should be taken to avoid making contact with other abdominal contents. The caudodorsal sac of the rumen is situated in the left caudodorsal abdomen, the proximity to the cervix varying according to rumen fill. This is the main part of the gastrointestinal viscera to be avoided during the procedure.

### 2.2. Surgical Procedure

Rectal palpation was conducted to determine suitability for the procedure as described above. The vulva was cleaned, dried and disinfected in a routine manner. The tail of the cow was held away to the side for the duration of the procedure. A cylindrical container filled with disinfectant and secured to the crush enabled maintenance of the instruments in a hygienic state and close at hand for the operator [10].

A vaginal spreader was gently inserted into the vagina, the blades opened briefly to reduce resistance in the vaginal wall, and then removed. This instrument is similar to the Rice pelvimeter with rounded, atraumatic tips and edges to the blades. Its purpose is to permit forward passage and greater mobility of the surgeon’s hand in the vagina.

An incision was made in the vaginal fornix, dorsal to the cervix, on or slightly right of the midline. This was conducted by passing a Willis spay instrument (or ovariotome) [Bainbridge Pty Ltd., Luscombe, Australia] up the vagina to the fornix and then pushing forward with a firm but controlled thrusting movement. An alternative method is for the vaginal wall to be stretched with a curved tensioner and the incision made with a hand-held scalpel blade or hooked knife [14,21]. The vaginal incision was sufficiently deep to perforate the peritoneum in a single movement. The incision instrument was then removed.

A clean, lubricated hand was passed into the vagina. The incision was located and enlarged by blunt dissection to first allow one, then two fingers to enter the peritoneal cavity (Figure 2(1)). The forefinger and middle finger were passed through the incision and directed towards the ovaries. The flexibility of the vaginal wall and mobility of the cervix allowed the surgeon sufficient movement to explore the pelvic and caudodorsal abdominal cavities. Under digital guidance, the ovary was located and the ovarian bursa identified on the craniolateral aspect.

Once located, the ovarian bursa was clasped between the fingers as shown in Figure 1 and drawn upwards towards the vaginal incision (Figure 2(2)). The operator then inserted two fingers into the ovarian bursa to isolate the mesosalpinx and uterine tube (Figure 2(3)). Care was taken to avoid the bursa tearing, which would render the identification of the structures by digital palpation impossible and was reason to abort the procedure.

The uterine tube cutting tool (Figure 3) consisted of two inward-facing size 22 scalpel blades mounted in a slot in a protected head assembly, with a shaft approximately 400 mm in length. The blades were spaced close to each other such that they severed tissue pulled into the instrument’s slot. The cutting instrument was passed into the vagina, through the incision, toward the ovary. The ovarian bursa was held under tension and the blades of the cutting tool passed around the fold of bursa containing the uterine tube (Figure 2(4)). The cutting tool was pushed forward and the bursa pulled back between the blades to sever a segment.

The tool and hand were removed holding the fold of ovarian bursa, which was inspected to confirm the presence of the uterine tube (Figure 4). Typically, a section of tube several centimetres in length was removed.

The procedure was repeated for the contralateral ovary and no attempt was made to close the vaginal wounds.

Dropped ovary spaying of heifers was performed by introducing a Willis spay instrument into the peritoneal cavity as described above, locating each ovary in turn using rectal palpation and passing it through the eye of the cutting instrument, transecting the ovarian attachments and dropping the ovaries into the peritoneal cavity.

Animals were given immediate access to feed and water and further management was as described above.

## 3. Results

The procedures were performed as described with no complications. The reproductive status of each animal was not recorded but a high proportion of cows and some of the heifers were pregnant with many beyond the first trimester of gestation.

The descriptive results for the duration of the procedures and surgeries, morbidity and weights are shown in Table 1. No mortalities were experienced in this study. One cow was diagnosed with pneumonia the day after surgery while another was diagnosed with metritis four days after surgery. Both of these animals were treated and withdrawn from the study. A number of animals showed signs consistent with recovery from pelvic surgery for a variable time after surgery, namely raised tails, muco-sanguinous vulvar discharge, tenesmus, arched backs and reduced mobility. No animals were seen to abort pregnancies and no placental or foetal tissues were found in the pen. No animals were noted to be showing signs of ill health at the end of the study and all were released to pasture.

Most (79%) of the cows gained weight in the 9–10 days of the study. The amount of weight gain was similar to that of the eight control animals, which gained an average of 8 kg each. The spayed heifers mostly lost weight with a similar loss to that of the eight unspayed hiefers, which lost an average of 4.6 kg. The reasons for the incomplete weight data for one cow and one heifer are not known.

## 4. Discussion

A novel surgical method utilising the resection of the mesosalpinx and uterine tube described here provides an alternative to ovariectomy that is particularly useful for mature and pregnant cows in rangeland production systems. This procedure evolved in the context of the unique beef production systems in northern Australia. The associated breeding practices and welfare considerations were described by Holroyd and Petherick, amongst others [3,6].

In contrast to dropped ovary spaying, the incidence of postoperative complications and welfare outcomes following uterine tube resection had not been documented. No mortalities or signs of severe ill health were noted with either method. Signs of discomfort, such as straining, were noted despite the administration of a long-acting non-steroidal anti-inflammatory drug as a part of the procedure. Straining had been reported to cause eventration of the intestines through the vagina, which could be reduced under epidural anaesthesia if identified promptly [14]. Sepsis and peritonitis had been observed after uterine tube resection. A further study of behavioural indicators of pain associated with the procedure is underway.

No signs consistent with the interruption of pregnancies were noted despite a number of the animals in this study being far enough into gestation for us to expect some visible signs had they aborted. Given the location and remote facilities, the possibility that pregnancy losses might have occurred and were unnoticed could not be ruled out. The sequelae of the UTR in pregnant animals were not documented, although given the maintenance of normal ovarian function, it was expected to not adversely affect the maintenance and successful completion of gestation.

The continued displaying of behavioural signs of oestrus could potentially result in difficulty distinguishing animals that have undergone UTR from others in the herd. The visualisation of the segments of both uterine tubes and the clear identification of the animal was therefore important, as was accomplished in this study using permanent ear markings.

The weight gain observed in most of the cows in this study was surprising in light of the somewhat more invasive nature of UTR compared to spaying. It was not possible for the effect of the surgeries to be separated from the effects of other aspects of the procedure, notably the mustering, mixing of groups and holding of large numbers of animals in close proximity, proximity to humans, restraint in a crush, rectal palpation, administration of medications and placing of the ear tags and punches. We speculate that the heifers were likely to have been more negatively impacted by these procedures than the cows given their lack of any prior experience of these procedures.

Uterine tube resection is more technically demanding than the dropped ovary method of spaying in that it requires more advanced manipulative skills, additional specialised instrumentation and more detailed knowledge of the anatomy. The speed of the execution of both procedures noted in this study enables large numbers of animals to be processed in a reasonable time, thus making the procedures economically feasible.

Each of these methods of surgical contraception has relative advantages. Uterine tube resection involves the transection of smaller blood vessels, with a reported lower incidence of postoperative haemorrhage [6,14]. It may be preferable in animals further into gestation, since the success of complete removal of the ovaries with the dropped ovary technique is reduced beyond four months of gestation [10]. However, the vaginal distension and larger vaginal incision required for the UTR cause more apparent discomfort during and after the procedure, based on these limited qualitative observations.

Ovariectomy eliminates sexual behaviour when correctly performed, the exception being when remnants of ovarian tissue remain intact and functional [22]. The absence of oestrous behaviour is viewed as advantageous by some producers. It is also feasible in younger animals and is well tolerated due to the small size of the Willis instrument and the small surgical wound [13,22]. Nevertheless, spaying is a painful procedure and there is a risk of morbidity and mortality [13,15,22,23,24,25,26,27]. The main cause is postoperative haemorrhage arising from the ovarian blood vessels, which are not ligated. The incidence of complications is higher in cows and pregnant animals owing to the larger size of the ovarian vessels.

Making the incision is a critical point in any vaginotomy procedure including the UTR procedure described here. There is a risk of damage to non-target organs as the peritoneum is penetrated [10,11]. On the other hand, failure to enter the peritoneal cavity in the same motion as the incision of the vaginal wall is another potential pitfall, making it difficult to cut the peritoneum alone safely without creating retroperitoneal dead space, which in turn causes significant discomfort and predisposes the animal to postoperative complications.

The animals used in this study were well conditioned for the procedure as described in the methods. Animal health and welfare at the time of the procedure was an important factor in avoiding postoperative complications. Animals in very poor body condition or suffering from concurrent diseases may have higher incidences of morbidity and mortality after uterine tube resection or spaying.

The procedures are generally performed without sedation and requires the animals to be standing and restrained in a suitable crush. Although the procedure evidently induces stress, pain and inflammation (rectal palpation, tissue retraction and resection), routine analgesia is not mandated. Epidural anaesthesia may be recommended for analgesia but is not generally feasible or cost-effective given the high-throughput nature of the procedures in the Australian context.

Non-steroidal anti-inflammatory agents are recommended although not mandated by Australian animal welfare standards [17]. The use of analgaesia for invasive husbandry practices is expanding, with the Australian industry reporting that 84% of the cattle spayed in 2020 received pain relief [28].

The *Bos indicus* breeds may lend themselves to UTR due to anatomical features of the cervix, vagina and perineum. These breeds suffer from a higher incidence of cervico-vaginal prolapse, thought to stem from heritable differences in the pelvic structures [29,30]. These same factors may favour surgical manipulation per vaginum with greater ease than in other breeds.

Given that rangeland cattle are often unhabituated to humans, the process of mustering, being held in yards, and handling associated with rectal palpation and surgical spaying is likely to be stressful. The welfare impacts of the pain, stress and inflammation of this procedure are important and there may be differences in the impact of the procedure between heifers and cows. Thus, more research on the mitigation of pain for spay procedures is urgently required.

Ovariectomy is used to extend the persistence of lactation in dairy cows [31] and as a model for reproductive endocrinology [32]. The UTR method can be considered in other contexts where the establishment of pregnancy is undesirable but removal of the ovaries is not required or where the separation of the physical connection of the uterine tube between the ovaries and the uterus is indicated.

## 5. Conclusions

The short-term health and welfare outcomes following UTR are comparable to those of surgical spaying. Uterine tube resection performed via an incision in the vagina provides a viable alternative to ovariectomy for the surgical contraception of mature female cattle. Further research of the health and welfare implications is indicated.

## Figures and Tables

**Figure 1 animals-13-01066-f001:**
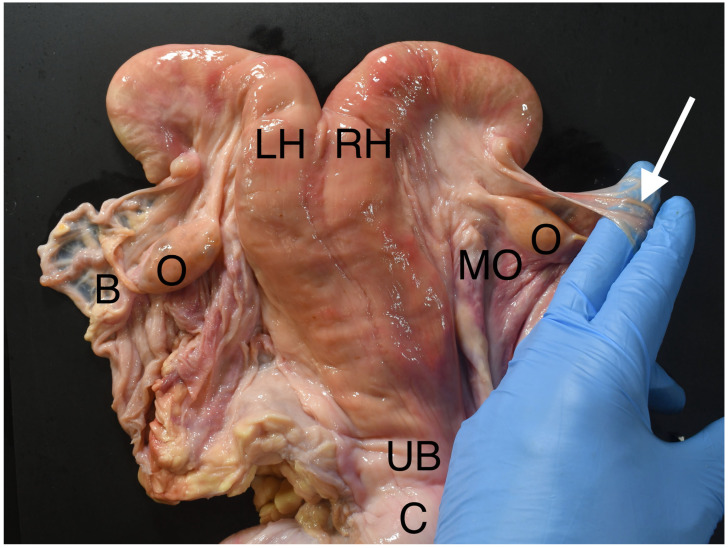
The uterus, ovaries and ligaments. C—Cervix; UB—Uterine body; LH, RH—Left and right uterine horns; O—Ovaries; MO—Mesovarium; B—Ovarian bursa. The arrow shows the uterine tube running over a finger inserted below the mesosalpinx into the right ovarian bursa. The positioning of the fingers demonstrates the grasping of the mesovarium during the surgical procedure.

**Figure 2 animals-13-01066-f002:**
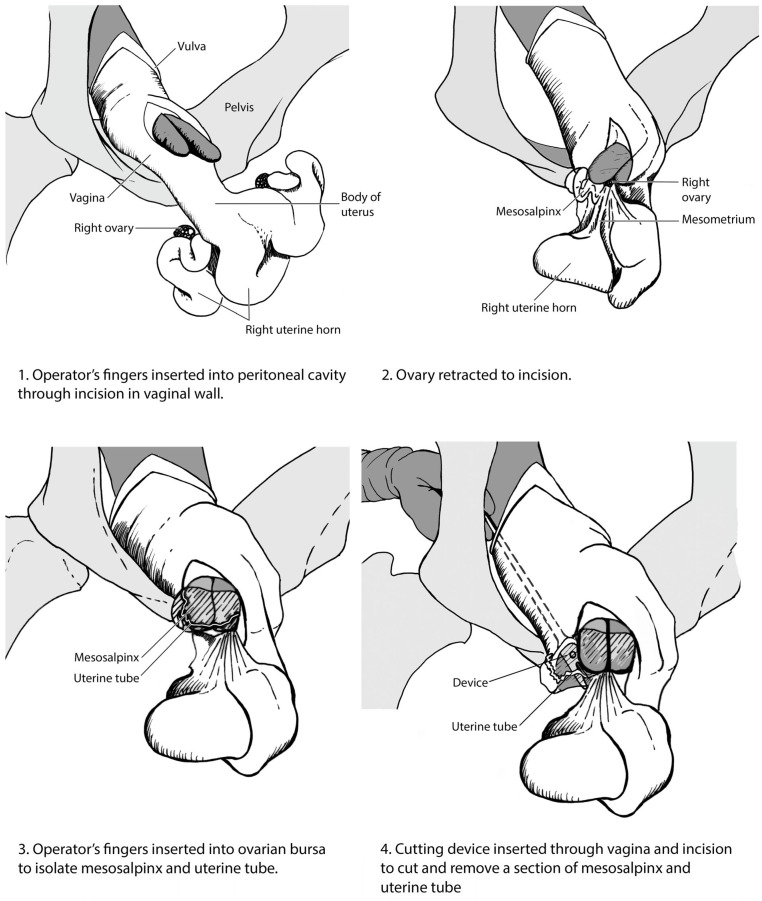
Stages of the uterine tube resection via the vaginal procedure as described in the text.

**Figure 3 animals-13-01066-f003:**
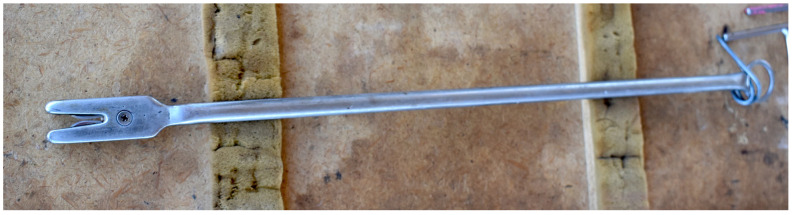
The custom-made uterine tube cutting tool.

**Figure 4 animals-13-01066-f004:**
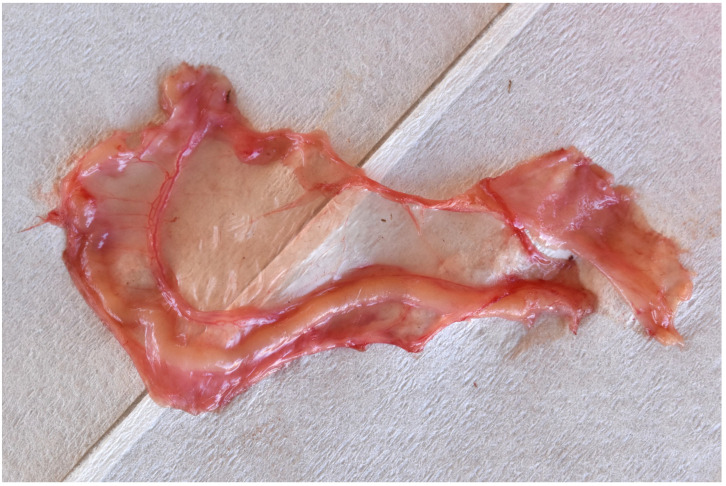
A section of mesosalpinx and uterine tube removed per vaginum.

**Table 1 animals-13-01066-t001:** Numbers of animals, procedure times, morbidity and weight of cows subjected to UTR and heifers spayed using the dropped ovary technique.

	Uterine Tube Resection	Spays
Number of animals	70	60
Duration of procedure from rectal palpation to end of surgery: Average (range)	3 min 5 s (1 m 43 s–5 min) ^1^	49 s (21 s–2 min 39 s) ^2^
Duration of surgery: Average (range)	1 min 13 s (23 s–2 min 6 s) ^1^	20 s (6 s–1 min 43 s) ^2^
Post-procedure morbidity	2 animals	0
Weights at start: Average (±SD)	414.5 kg (±47.4)	231.9 kg (±26.1)
Weights at start: Range	314 to 534 kg	169 to 284 kg
Weight change: Average (±SD)	+9.3 kg (±14.5) ^3^	−3.5 kg (±13.3) ^4^
Weight change: Range	−63 to +36 kg	−35 to +22 kg
Number (%) of animals losing weight	13 (19%) ^3^	37 (63%) ^4^

^1^ Procedure times recorded for 31 cows and surgery times for 30. ^2^ Procedure times recorded for 28 heifers and surgery times for 27. ^3^ Complete weight records available for 67 cows. ^4^ Complete weight records available for 59 heifers.

## Data Availability

Data available on request.

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
