# Peer review of "Uterine Tube Resection by Vaginotomy as an Alternative to Ovariectomy in Mature Cattle"

_animals, 2023, doi:10.3390/ani13061066_

Round 1

Reviewer 1 Report

Reviewer Report on the paper titled:

Uterine tube resection by vaginotomy as an alternative to ovariectomy in mature cattle.

Animals ID – 2258365

General comments:

This work describes a technique for uterine tube resection (UTR) via colpotomy and compares the outcomes from 70 Brahman cows subjected to the procedure in comparison with 60 heifers spayed by the Dropped ovary technique. This subject matter is suitable for the journal Animals. Writing style, grammar and spelling are all of a high standard.

Specific comments:

 Simple Summary and Abstract

These are both well written, reflecting the content of the paper. I have just two points for consideration by the authors:

  • The anatomical term “Fallopian” tube is a human descriptor. The alternative descriptor of uterine tube for animals is “oviduct”.
  • The authors may wish to check historical references, but in the 1990’s, there was strong anecdotal evidence in (if I remember correctly) the Journal of Animal Science, that Dr Willis may have misrepresented himself as the developer of this technique. There was conjecture as to who actually developed this technique and the recommendation was to simply describe the technique as the Dropped Ovary Technique.

 Introduction

The Introduction is well set out, provides good background justification for the research and describes the scope of the paper.

 Materials and Methods

Generally well written and well described.

Line 122 – into

Line 134 – external ostium (os is a colloquialism). …surrounded by the vaginal fornix…

Line 154/55 – I’d like to see this sentence rearranged to reduce ambiguity - The mesosalpinx is a separate fold of the broad ligament holding the uterine tube, which arises at the tubal pole of the ovary and runs the length of the uterine tube between the ovary and the uterine horn.

Perhaps change to - The mesosalpinx is a separate fold of the broad ligament arising at the tubal pole of the ovary and running the length of the uterine tube, supporting it between the ovary and the uterine horn.

Line 185 – change vaginal to vagina

It is noted that there is Discussion on the absence of pregnancy loss in the trial, but no quantification of pregnancy percentages, or durations in the M&M’s

Results

Well presented. The one concern is that about 20% of the Results (6 lines of text) relate to pregnancy and subsequent maintenance. But there was no quantification of pregnancy percentage, or durations? Lines 106/107 suggest this was performed, but perhaps cows were not individually identified to allow the status to be recorded. If available, the percentage of the study animals that were pregnant and the durations would add value to the paper. As it stands, there seems little value in the comment on lines 236/7, as not much outward sign of loss would be expected if most pregnancies were eg 8 to 10 weeks. Documenting pregnancy duration would also allow the reader to gauge if pregnancy status affected the duration of the procedure.

Discussion

Generally well written.

Similar comment as with Results – Lines 262 to 266 discuss something that was not clearly documented or assessed in the paper. There may be value in leaving this in as is, but there would be more value if the pregnancy data could be provided.

Line 266 – “…not generally impact…” is awkward. Perhaps – ….expected to not adversely affect the maintenance….

Line 320 – a comma between “humans” and “the”.

References

References appear relevant, comprehensive and up-to-date.

Recommendation

This paper is well-written and presented. It provides useful information on the novel topic of UTR as a means of contraception in extensively managed beef cattle. I encourage the authors to carefully consider the information surrounding pregnancy status prior to final submission.

Reviewer 2 Report

General comments:

Please also describe (even if shortly) the Willis dropped ovary technique in the materials and methods session. While the Willis technique is commonly used, the common reader could be unfamiliar with the technique, and will thus lack important information to understand what the new (UTR) technique proposes. I believe it would be nice to emphasize the steroid production post surgery in animals exposed to each technique.

I think it would be nice for you to expand your discussion on post-operative steroid production between the two methods. Seems like the UTR technique would allow for progression of gestation in case the female exposed to surgery was pregnant, while the Willis technique would likely terminate pregnancy. Further, it seems like the Willis technique would be a more "convincing" technique to the naïve producer because females that underwent spaying by the Willis technique will not show signs of estrus, while females exposed to the UTR would. In other words, in the Willis technique, it is easy to confirm the effectiveness of the surgery by simply observing animal behavior and confirming lack of estrus activity. While if the UTR technique is performed by an unexperienced veterinarian who is unsure of the effectiveness of the surgery, there would be no way to confirm the success of the surgery as animals that underwent a successful surgery or an unsuccessful surgery would have the same behavior, and all palpable structures intact. How would you expect these characteristics of the UTR technique to influence its adoption? 

You do mention that the small size of vagina in heifers makes it difficult for the UTR technique to be performed, and that is the reason for which you used cows for that group. Why did you choose heifers as your Willis control instead of cows? This category effect is a confusing factor when one analyzes the performance data post surgery. The reasons for choosing heifers rather than cows as control should be explained.

Specific comments:

Line 19: Please use oviduct, as it is the correct term in the bovine. I realize this is used in the simple summary session and this session is meant to be more comprehensive to the general public. Perhaps say: "removing part of the uterine tubes (the equivalent to the fallopian tubes in humans)" 

Lines 44-46: I wouldn't start your introduction with this sentence. Maybe move it to the materials and methods session where you describe the animal research ethics committee protocol.

Lines 101-102: You explain that for the UTR technique you need cows due to the size of the vagina being appropriate for the technique. Why did you choose to have heifers as control instead of cows?

Line 122: released into (missing “o”)

Line 198: As you say "once located the ovarian bursa was grasped between the fingers" I would suggest referring to figure 1 again.

Lines 206-207: Then I wouldn't mention it in the M&M session. If you wish to bring this up during the discussion, and comment on how it would have changed the procedure had it been available, that'd be more appropriate

Line 263: You say "being in calf" do you mean pregnant? please reword.
